# *TREM1* rs2234237 (Thr25Ser) Polymorphism in Patients with Cutaneous Leishmaniasis Caused by *Leishmania guyanensis*: A Case-Control Study in the State of Amazonas, Brazil

**DOI:** 10.3390/pathogens10040498

**Published:** 2021-04-20

**Authors:** José do Espírito Santo Júnior, Tirza Gabrielle Ramos de Mesquita, Luan Diego Oliveira da Silva, Felipe Jules de Araújo, Josué Lacerda de Souza, Thaís Carneiro de Lacerda, Lener Santos da Silva, Cláudio Marcello da Silveira Júnior, Krys Layane Guimarães Duarte Queiroz, Diogo Matos dos Santos, Cilana Chagas da Silva, Héctor David Graterol Sequera, Melissa Tamayo Hermida, Mara Lúcia Gomes de Souza, Marcus Vinitius de Farias Guerra, Rajendranath Ramasawmy

**Affiliations:** 1Programa de Pós-Graduação em Imunologia Básica e Aplicada, Universidade Federal do Amazonas, Manaus 69080-900, AM, Brazil; sdjunior.biol@gmail.com; 2Faculdade de Medicina Nilton Lins, Universidade Nilton Lins, Manaus 69058-030, AM, Brazil; luandiego.oliveira@gmail.com (L.D.O.d.S.); felipejules86@gmail.com (F.J.d.A.); josue.com@live.com (J.L.d.S.); thaisbelo3@gmail.com (T.C.d.L.); diogomsanttos@gmail.com (D.M.d.S.); mlss_th@hotmail.com (M.T.H.); 3Programa de Pós-Graduação em Medicina Tropical, Universidade do Estado do Amazonas, Manaus 69040-000, AM, Brazil; tirzagabi@gmail.com (T.G.R.d.M.); lener.santos77@gmail.com (L.S.d.S.); hector.graterol4@gmail.com (H.D.G.S.); maralgsouza@gmail.com (M.L.G.d.S.); mvfguerra@fmt.am.gov.br (M.V.d.F.G.); 4Fundação de Medicina Tropical Doutor Heitor Vieira Dourado, Manaus 69040-000, AM, Brazil; claudiomarc99@gmail.com (C.M.d.S.J.); kryslayane@gmail.com (K.L.G.D.Q.); cilanasilva15@gmail.com (C.C.d.S.); 5Genomic Health Surveillance Network: Optimization of Assistance and Research in the State of Amazonas (REGESAM), Manaus 69040-000, AM, Brazil

**Keywords:** triggering receptor expressed on myeloid cells-1 (TREM-1), polymorphism, cytokines, cutaneous leishmaniasis, *Leishmania guyanensis*, Amazonas

## Abstract

Background: Leishmaniasis is an infectious disease caused by *Leishmania* parasites. A Th1 immune response is necessary in the acute phase to control the pathogen. The triggering receptor expressed on myeloid cells (TREM)-1 is a potent amplifier of inflammation. Our aim is to identify whether the *TREM1* variant rs2234237 A/T (Thr25Ser) is associated with the disease development of cutaneous leishmaniasis (CL) in *Leishmania guyanensis*-infected individuals. The effects of the rs2234237 genotypes on plasma cytokines IL-1β, IL-6, IL-8, IL-10, MCP-1 and TNF-α are also investigated. Methods: 838 patients with CL and 818 healthy controls (HCs) living in the same endemic areas were genotyped by Polymerase Chain Reaction-Restriction Fragment Length Polymorphism. Plasma cytokines were assayed in 400 patients with CL and 400 HCs using the BioPlex assay. Results: The genotypes’ and alleles’ frequencies were similar in both patients with CL (AA = 618, 74%; AT = 202, 24%; TT = 18, 2%) and in HCs (AA = 580, 71%; AT = 220, 27%; TT = 18, 2%). Rs2234237 showed a modest effect on plasma IL-10 that disappeared when correction of the *p*-value was applied. Plasma IL-10 by rs2234237 genotypes were (mean ± SEM; AA = 2.91 pg/mL ± 0.14; AT = 2.35 pg/mL ± 0.12; TT = 3.14 pg/mL ± 0.56; *p* = 0.05). Conclusion: The *TREM1* rs2234237 (Thr25Ser) seems to have no influence on the susceptibility or resistance to *L. guyanensis* infections.

## 1. Introduction

*Leishmania* infections cause a broad spectrum of clinical manifestations. The clinical outcome depends on the *Leishmania* spp. and may range from asymptomatic, self-healing or non-healing skin lesions, classified as cutaneous leishmaniasis (CLs), to severe destructive mucosal lesions (MLs) or visceral leishmaniasis (VLs) that may even be fatal [1]. According to the World Health Organization (WHO), 97 countries were considered endemic for leishmaniasis in 2018, with 88 countries for CL, 78 for VL and 69 for both CL and VL [2]. In 2018, 253,435 new CL cases were reported worldwide [2]. In Brazil, 16,432 new cases of American Tegumentary Leishmaniasis (ATL) were reported in 2018 [3]. In the state of Amazonas, 1684 new cases of ATL were noted in 2018 [3].

The clinical outcome of *Leishmania* infections depends on a fine regulation of the immune response coordinated by T helper cells [4]. Th1 cells, producing high levels of IFN-γ, activate infected macrophages to induce superoxide production to contain the *Leishmania* parasite’s replication [4]. Alternatively, a Th2 response, producing interleukin-4 (IL-4), IL-5 and IL-13 favors the survival and replication of the *Leishmania* parasites [5].

Receptors expressed in innate immune cells play key roles in Th1/Th2 differentiation. A triggering receptor expressed on myeloid cells (TREM-1) is a receptor with a molecular mass of 30 kDa that belongs to the immunoglobulin-like superfamily. TREM-1 is highly expressed on activated monocytes/macrophages and neutrophils [6]. TREM-1 has a short intracellular tail without an immunoreceptor tyrosine-based activation motif (ITAM) domain. To activate its signaling pathway, TREM-1 forms a complex with the protein adapter DNAX-activating protein 12 (DAP12) for signal transduction [6,7]. The TREM-1/DAP12 complex signaling pathway activates phospholipase C γ (PLCγ), extracellular signal-regulated kinase (ERK) and phosphatidylinositol 3-kinase (PIK3) to trigger calcium mobilization, cell survival, neutrophil degranulation and increases in the expression of pro-inflammatory cytokines, resulting in hyper-inflammatory conditions [6,8].

Emerging evidence has shown the importance of TREM-1 in the modulation of an excessive pro-inflammatory response against *Escherichia coli* [7], Lipopolysaccharide-induced septic shock [7], *Pseudomonas aeruginosa* [9], *Aspergillus fumigatus* [10], Lymphocytic Choriomeningitis Virus (LCMV) [11], human immunodeficiency virus (HIV) [12] and *Mycobacterium tuberculosis* [13]. TREM-1 may be a promising therapeutic target in infectious diseases with exacerbation of inflammation.

Knockout mice of *TREM1* (*TREM1^−/−^*), upon infection with *L. major*, showed smaller lesions with reduced neutrophil infiltration after 14 days when compared to wild-type mice, despite both presenting similar parasitic load [14]. In addition, *TREM1^+/+^* neutrophils showed low Caspase3/7 activity compared to *TREM1^−/−^* neutrophils [14]. *CCL7*, *IL8*, *IFI44L* and *IL1B* genes were upregulated in *L. braziliensis*-stimulated peripheral blood mononuclear cells (PBMCs) from healthy high IFN-γ producing individuals [15]. The upregulation was cited to be related to the IL-17 and TREM-1 pathways, as determined by Ingenuity Pathway Analysis [15]. In *L. braziliensis*-infected patients with CL, miR-193b, miR-671 and TREM-1 were shown to be negatively correlated with good responses to treatment and faster wound healing [16]. TREM-1 also performed cross-talk with toll-like receptors 2 (TLR2) and TLR4 pathways to potentiate the intracellular TLR signal and upregulate cytokines’ expression [17,18,19,20].

Variants present in the *TREM1* gene are associated with susceptibility to inflammatory bowel diseases [21], sepsis prognosis [22], pneumonia [23], coronary artery disease [24], septic shock [25], severe malaria [26] and atherosclerosis [27]. The variant rs2234237 A/T is a non-synonymous variant present in Exon 2 of the *TREM1* gene. This variant results from the substitution of threonine with serine at the 25th amino acid position in the extracellular immunoglobulin-like domain, responsible for binding with TREM-1 ligands. Furthermore, this variant is correlated with an increased level of soluble TREM-1 [25].

Taking into account the importance of TREM-1 in inflammatory diseases, this study investigates whether rs2234237 A/T is a risk factor in the development of CL in patients infected by *L. guyanensis*. We also analyze rs2234237 A/T’s effects on circulating cytokines’ plasma levels.

## 2. Results

### 2.1. Characteristics of the Population

The characteristics of the study population are described elsewhere [28]. Briefly, in this study, the population was composed of 838 patients with CL and 818 HCs. The main characteristics of the participants of the study are shown in Table 1. Females were more frequent among the control subjects (*p*-value = 0.002). The controls were older than the patients with CL (*p*-value < 0.001). All patients with CL included in the study had fewer or equal to (≤) six skin lesions. Patients with HIV and pregnant women were excluded. The diameters of the lesions were not recorded as all patients sought treatment only after the lesions were well developed. Besides this, most patients applied local traditional treatment before going to the hospital.

### 2.2. Genotypes and Allele Frequencies of the TREM1 Variant rs2234237 A/T’s Polymorphism

The genotype distributions did not present any HWE deviation in either patients with CL or HCs. The genotypes’ and alleles’ frequencies of rs2234237 were similar in both patients with CL (AA: 74%, n = 618; AT: 24%, n = 202; TT: 2%, n = 18) and the healthy controls (AA: 71%, n = 580; AT: 27%, n = 220; TT: 2%, n = 18), as shown in Table 2.

Genotypes’ and alleles’ comparisons did not show any statistical differences between the patients with CL and the HCs despite the frequency of T alleles being slightly higher (16%) among the HCs when compared with the patients with CL (14%). These results show that the polymorphism of the variant rs2234237 A/T (Thr25Ser) in the TREM1 gene is neither associated with the development of CL in patients infected by *L. guyanensis* nor with resistance.

### 2.3. Effects of TREM1 rs2234237’s Polymorphism on Circulating Plasma Cytokine Levels

The TREM-1 signaling pathway’s activation induces the expression of the pro-inflammatory cytokines IL-1β, IL-6, IL-8, MCP-1 and TNF-α and downregulates IL-10 expression [6,14]. The effect of the variant rs2234237 (Thr25Ser) on plasma cytokines was analyzed in 800 subjects (400 patients with CL and 400 HCs). The means and standard of the means in pg/mL (mean ± SEM pg/mL) of plasma cytokines, adjusted by sex and age, are shown for patients with CL, for HCs and for the total individuals (patients with CL + HCs) in Appendix A. The means ± the SEM pg/mL of each cytokine for the rs2234237 genotypes and inheritance models (codominant, dominant, recessive and over-dominant) are shown in Appendix A. Of all of the cytokines analyzed, only IL-10 seems to be influenced by the variant rs2234237 and the effect is modest. The distribution of the circulating plasma levels of IL-10, as shown in Figure 1, is different among the genotypes in the combined samples (patients with CL + HCs) (*p* = 0.05). Among carriers of the AA genotype, the mean circulating levels of IL10 are different (2.91 ± 0.14 pg/mL, n = 492) compared to the AT (2.35 ± 0.12 pg/mL, n = 174) and TT genotypes (3.14 ± 0.56 pg/mL, n = 15). In the dominant model, individuals bearing a T allele (AT+TT genotypes, n = 189) had a mean level of circulating plasma IL-10 of 2.41 ± 0.12 pg/mL compared to 2.91 ± 0.14 pg/mL for individuals homozygous for the A allele (AA genotype, n = 492; *p* = 0.03). However, in a recessive model, individuals homozygous for the T allele seemed to have higher levels of IL-10 (3.14 ± 0.56 pg/mL, n = 15) compared to individuals carrying the A allele (AA+AT: 2.76 ± 0.10 pg/mL, n = 666, *p* = 0.59). Interestingly, in the over-dominant model, heterozygous individuals seemed to produce lower levels of circulating plasma IL-10 (AT: 2.35 ± 0.12 pg/mL, n = 174) when compared with both sets of homozygous individuals (AA+TT: 2.91 ± 0.13 pg/mL, n = 507, *p* = 0.02). Furthermore, we did not observe a difference between the two sets of homozygous individuals (AA: 2.91 ± 0.14 pg/mL, n = 492; TT: 3.14 ± 0.56, n = 15; *p* = 0.76). Of note, the statistical significance disappeared when the Benjamini and Hochberg correction was applied, assuming five tests, as depicted by the *p*-value.

## 3. Discussion

In the area of endemicity for *Leishmania* species, some individuals develop clinical symptoms while others sharing the same environment and exposure to the parasite do not show any symptoms or remain asymptomatic. The clinical spectrum of leishmaniasis also suggests that hosts’ genetic factors are involved in the disease outcome. This points to possible host genetic contributions to the development of leishmaniasis [28,29,30]. Understanding the role of host genetics in *Leishmania* infections may decipher the developmental mechanism of the disease and contribute to elaborating a novel treatment for leishmaniasis, especially in the field of immunotherapy.

In this study, our data showed that the variant rs2234237 (Thr25Ser) within the *TREM1* gene in Exon 2 was not associated with the development of CL or resistance to *L. guyanensis* infection. In contrast, the variant may have influenced the expression of IL-10, as the T allele correlated with lower levels of circulating plasma IL-10, although the statistical significance disappeared when corrections were applied for multiple comparisons.

A few studies have shown that the rs2234237 T allele is associated with several diseases. The T-allele has been suggested to be a risk factor for the development of severe malaria in African children [26] and Behcet’s disease in the Korean population [21]. The T-allele has also been reported to be associated with sepsis in Chinese patients [22] and high risk of developing pneumonia caused by mechanic ventilation [23].

Conversely, other studies have reported a lack of association of rs2234237 with septic shock [25], severe sepsis [31], or a clinical course of sepsis [32]. One study showed that individuals homozygous for the rs2234237 A allele were at a high risk of developing coronary artery disease (CAD) among the Russian population [24].

The TREM-1 signaling pathway acts synergistically with TLR2 and TLR4. This leads to NF-kB activation and expressions of proinflammatory cytokines [17,18,19,20]. *Plasmodium berghei*-infected inbred mice treated with recombinant mouse TREM-1/Fc and mouse TREM-1/antibody to inhibit TREM-1 activity exhibited a sharp decrease in TNF-α, IL-6, IFN-γ and IL-10 levels and less tissue damage of affected organs, compared with untreated mice [33]. Whole blood from patients with simple sepsis showed a higher expression of TREM-1 and higher levels of IL-6, IL-8, IL-10 and TNF-α upon stimulation with LPS when compared with whole blood from patients with severe sepsis [20]. Pathway-specific microarray analysis revealed that the silencing of TREM-1 induced a reduction in the expression of myeloid differentiation protein (MyD88) (a protein adapter of TLRs), IL-10 and IL-1β [34]. In contrast, other studies have shown that TREM-1 can inhibit IL-10 expression [10,35]. Human peripheral monocytes from *L. braziliensis*-infected patients with CL exhibited higher expression of TLR2, TLR4, TNF-α and IL-10 upon infection in vitro with *L. braziliensis* [36].

In this study, the effects of rs2234237 on the levels of plasma circulating IL-1B, IL-6, IL-8, IL-10, TNF-α and MCP-1 were analyzed. Only plasma IL-10 seems to be influenced by the rs2234237 genotypes, as shown in Appendix A. The AA homozygotes were correlated with higher circulating IL-10 compared with the T genotypes (AT + TT). Of note, as the frequency of the T allele was low in our population, only 15 individuals had the TT genotypes out of the 666 individuals assayed for IL-10. When compared, the levels of circulating plasma IL-10 between the AA (2.91 ± 0.14 pg/mL) and TT (3.14 ± 0.56 pg/mL) homozygotes revealed that TT genotypes correlated with high levels of IL-10. However, the standard error of the mean among the TT genotypes was higher (0.56) when compared to 0.14 for the AA genotypes. The T allele may, in fact, correlate with low levels of IL-10.

Comparisons of genotypes’ frequencies did not reveal any statistical differences between the patients with CL and the HCs, but pointed out that carriers of the T allele may have an 18% risk of developing CL in a dominant model (OR = 1.18 (95% CI [0.95–1.47])). In a study of falciparum malaria with 293 and 87 African children with uncomplicated and severe malaria, respectively, individuals with the T allele had an increased risk of developing severe malaria (dominant model; OR = 2.4 (95%CI [1.2–4.5]) [26]). Likewise, under the dominant model, the T allele was associated with Behcet‘s disease (OR = 1.69 (95%CI [1.1–2.6])) [21]. Although no statistical significance has been observed in our analysis, we can hypothesize that individuals with the T allele may not properly control the inflammation and thus progress to the development of a lesion. The rs2234237 T allele was cited to correlate with an increased level of soluble TREM-1 in human whole blood stimulated with LPS [25]. Soluble TREM-1 (sTREM-1) may act as a decoy receptor for TREM-1 ligands and lead to a decrease of circulating IL-10. Interestingly, TREM-1 silencing lowered the expression of IL-10 [34]. In contrast, sTREM-1 was suggested as a possible modulator of the anti-inflammatory pathway [37].

This study has several limitations, especially with the correlation of plasma IL-10 by the rs223437 genotypes. The frequency of the T allele (16%) is low when compared with the A allele (84%). Of 1656 individuals’ genotypes, only 36 were homozygous for the T allele. Of these, only 15 were assayed for cytokines. To show that the T allele may be correlated with low plasma IL-10, we had to pool healthy controls and patients with CL together to have statistical power. The pooling was performed as we were looking at whether the allele had an influence on expression. Age and gender did not influence plasma IL-10, as shown by linear regression correlation analysis (Appendix A). Future studies are needed to confirm this correlation as it is very modest and it might be a spurious association. It would be interesting in future studies to select only 15 individuals from each genotype and perform stimulation studies with whole blood in vitro to avoid confounding factors.

## 4. Materials and Methods

### 4.1. Population Studied

In this study, individuals were recruited from the perirural regions of Manaus, the capital city of the Amazonas state, considered endemic for *L. guyanensis*. The recruitment of patients with CL and healthy controls (HCs) (individuals with no history of CL or any scars and different lesions) from the endemic areas was initiated in January 2009 and ended in March 2016. Patients with CL attended the Fundação de Medicina Tropical Doutor Heitor Vieira Dourado (FMT-HVD), a referral hospital, for the treatment of tropical diseases including leishmaniasis. The majority of the participants in the study are agricultural or farmworkers. The population studied has a mixture of Native American (50 to 60%), European (40 to 50%), and African (around 10%) ancestries [38].

### 4.2. Ethics Statement

This study was conducted in accordance with the international Declaration of Helsinki and was approved by the Research Ethics Committee of the Fundação de Medicina Tropical Dr. Heitor Vieira Dourado (FMT-HVD), granted under the file number CAAE:09995212.0.0000.0005 on 31 May 2013. All of the participants (or their responsible party for individuals less than 18 years old) provided written informed consent for the collection of biological samples and their subsequent analysis.

### 4.3. Collection of Biological Samples

Five mL of peripheral blood from each participant in the study was collected using venipuncture and transferred into an EDTA-containing vacutainer tube (Becton Dickinson) for DNA extraction and plasma separation for cytokines’ assay. A skin biopsy sample around the lesions from all patients with CL was collected for the identification of *Leishmania* species.

### 4.4. Leishmania Genotyping

The presence of *Leishmania* in the skin lesions of patients with CL was confirmed by direct microscopic examination of Giemsa-stained scarifications of the lesions. DNA was extracted from biopsy lesions of all patients with CL. The discrimination of *L. Viannia* from the *L. Leishmania* subgenus was performed by polymerase chain reaction (PCR) as described elsewhere [39,40]. The identification of *Leishmania* species was obtained by direct nucleotide sequencing of heat-shock protein 70 (HSP70) and miniexons genes as described previously [29,30].

### 4.5. Cytokines’ Assay

At the time of enrolment in the study, all participants provided 5 mL of peripheral blood. All patients with CL were treatment naïve prior to collection of the blood. Plasma was separated from the peripheral blood of both patients with CL and the HCs, and all samples were kept frozen at −80 °C for the cytokines’ analyses. Concentrations of circulating cytokines IL-1β, IL-6, IL-8, IL-10, monocyte chemoattractant protein-1 (MCP-1) and tumor necrosis factor-α (TNF-α) were detected from the plasma using the multiplex cytokine commercial kit Bio-PlexProTM Human Cytokine Grp I Panel 27-Plex (Bio-Rad, Hercules, CA, USA), according to the manufacturer’s instructions, in the Bio-plex 200 Protein Array System (Luminex Corporation, Austin, TX, USA).

### 4.6. TREM1 Polymorphism Genotyping

DNA was extracted using proteinase K and the salting-out method. Allele identification was performed by polymerase chain reaction-restriction fragment length polymorphism (PCR-RFLP). The following pair of primers, TREM1 Forward: 5′ TCT TTC CCT GCT TAT AGA ACT CCG 3′ and TREM1 Reverse: 5′ GGT CTT GGG CAT CTC TCC GTC CTT 3′, was designed to flank the polymorphism region to generate a fragment of 166 base pairs (bp) by PCR in a final volume of 25 µL containing 50 ng of genomic DNA, 0.2 pmol/L of each primer, 40µmol/L of each deoxynucleotide triphosphate (dNTP), 1 U of Taq polymerase, 2.0 mmol/L of MgCl2 in a buffer containing 500 mmol/L KCl and 100 mol/L of Tris-HCL (pH 8.3). The PCR reactions were optimized in the Mastercycler ep S (Eppendorf) under the following cycling conditions: initial denaturation step at 95 °C for 5 min; 45 cycles of 95 °C for 15 s, 56 °C for 15 s and 72 °C for 30 s; and a final step of 72 °C for 7 min. A volume of 10 µL of the PCR product was digested using two units of *Mse*I enzyme restriction (New England Biolabs). The DNA fragments were size separated by electrophoresis in a 3% agarose gel stained with 1 µL of 10 mg/mL ethidium bromide per 100 mL of gel. In the presence of the A allele, two fragments of 128bp and 38bp were generated. When the T allele was present, the fragment of 166pb remained uncleaved.

### 4.7. Statistical Analyses

The Mann-Whitney and χ^2^ tests were used to compare the ages and sexes in both the CL and control groups. The allelic and genotypic frequencies were determined by direct counting. Deviations in the Hardy–Weinberg Equilibrium (HWE) were tested comparing the observed genotype frequencies to the expected genotype frequencies. Association analyses were performed by logistic regression using the statistical language R (http://www.r-project.org/ accessed on the 20 September 2020). Single nucleotide polymorphism (SNP) association to disease outcome and SNP influence on circulating plasma cytokines were tested using the SNPassoc library version 2.0-0 [41]. Multiple testing corrections were realized according to Benjamini and Hochberg’s method [42] to control for false discovery rates using the R stats library. To control for false-positives relating to the influence of the SNP on circulating cytokines, the tail area-based false discovery rate values (FDR *q*-values) using the R package fdrtool were applied [43]. Cytokines plots were designed with the ggplot2 package version 3.3.6 [44].

## 5. Conclusions

Altogether, our data showed that the variant rs2234237 of the TREM1 gene is not associated with the development of CL in patients infected with *L. guyanensis*. Further studies including other variants present in TREM1 gene are necessary to better understand the role of this receptor in the immune system and modulation of cytokine expression in response to *Leishmania*-infections.

## Figures and Tables

**Figure 1 pathogens-10-00498-f001:**
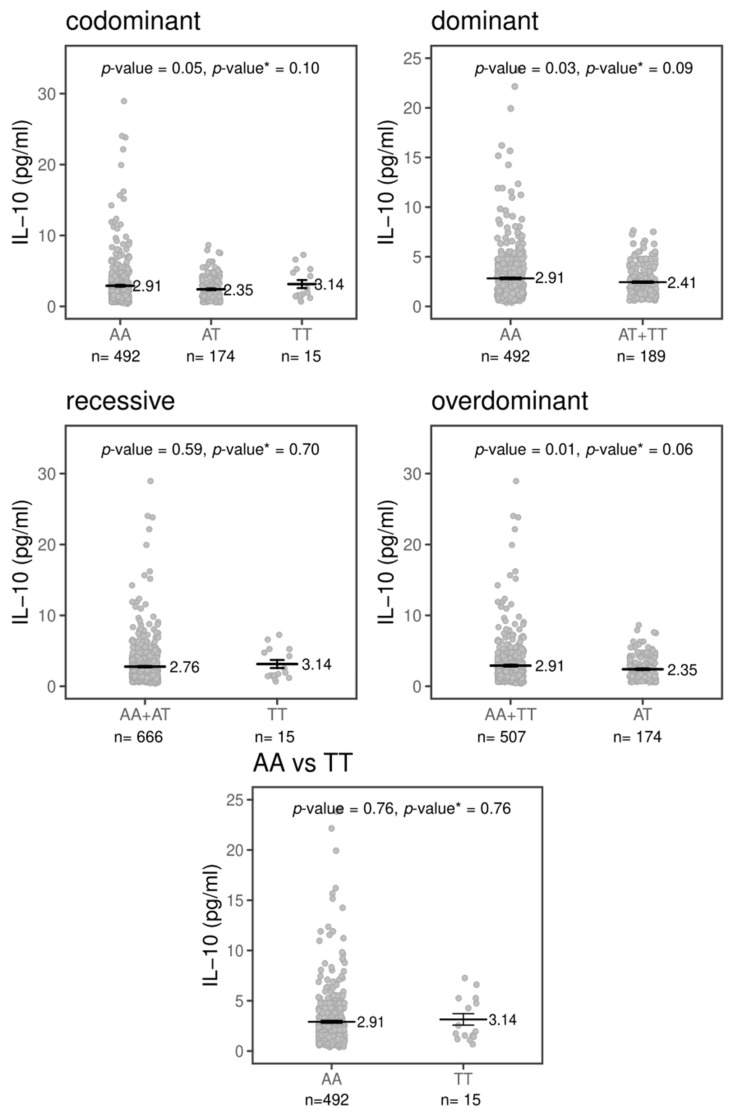
*TREM1* rs2234237 A/T (Thr25Ser) influence on circulating IL-10 plasma levels in total subjects in four inheritance models (codominant, dominant, recessive and over-dominant) and between homozygous genotypes adjusted by sex and age. The black bars represent the means of concentration in picogram per milliliter (pg/mL), whereas error bars represent the standard error of mean. *p*-value <0.05 is considered significant. *p*-value*: Benjamini and Hochberg’s correction assuming five tests.

**Table 1 pathogens-10-00498-t001:** Basic characteristics of the study population.

	Patients with CL		HCs ^1^		
	*N* = 838		*N* = 818		
	Males	Females	Males	Females	*p*-Value ^2^
Sex	629 (75%)	210 (25%)	567 (68.2%)	265 (31.8%)	0.002
Age (mean ± SEM ^3^)	34.04 ± 0.54	37.12 ± 1.09	42.02 ± 0.72	40.05 ± 1.14	<0.001

^1^ Healthy controls. ^2^
*p*-value < 0.05 is significant. ^3^ Standard error of mean.

**Table 2 pathogens-10-00498-t002:** Genotypes’ and alleles’ frequencies for rs2234237 (Thr25Ser) polymorphism in patients with cutaneous leishmaniasis and healthy controls, and statistical comparisons adjusted by sex and age.

Genotype and Allele Frequencies				
	Patients with CL		HCs ^1^	
rs2234237	Total = 838		Total = 818	
Genotypes				
AA	618 (74%)		580 (71%)	
AT	202 (24%)		220 (27%)	
TT	18 (2%)		18 (2%)	
Alleles				
A	1.438 (86%)		1.380 (84%)	
T	238 (14%)		256 (16%)	
Statistical comparisons between patients with CL and HCs ^1^				
Inheritance models	OR ^2^ [CI ^3^ 95%]	*p*-value ^4^	Corrected *p*-value ^5^	*q*-value ^6^
Codominant				
AA				
AT	1.19 [0.95–1.50]	0.31	0.39	0.12
TT				
Dominant				
AA vs. AT+TT	1.18 [0.95–1.47]	0.14	0.32	0.09
Recessive				
AA+AT vs. TT	0.99 [0.50–1.94]	0.97	0.97	0.29
Over-dominant				
AA+TT vs. AT	1.19 [0.95–1.50]	0.13	0.32	0.09
Log-additive	1.14 [0.93–1.39]	0.19	0.32	0.09

^1^ Healthy control. ^2^ Odds ratio. ^3^ Confidence intervals. ^4^
*p*-value uncorrected. ^5^ Corrected *p*-value: Benjamini and Hochberg’s correction assuming five tests. ^6^
*q*-value: FDR (false discovery rate).

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
