# Peer review of "TREM1 rs2234237 (Thr25Ser) Polymorphism in Patients with Cutaneous Leishmaniasis Caused by Leishmania guyanensis: A Case-Control Study in the State of Amazonas, Brazil"

_pathogens, 2021, doi:10.3390/pathogens10040498_

Round 1
Reviewer 1 Report
This article importantly finds no difference in TREM-1 genotypes in patients with L. guyanensis compared to healthy controls, however the details of the populations and how well matched they are is not clear. Furthermore, the secondary aim of looking at cytokine levels inappropriately pools healthy controls and infected patients for final analysis. The surprising finding of no difference in cytokine levels (except a possible difference in IL-10 which disappears with more rigorous statistical analysis) in patients with separate polymorphisms goes against the literature, and the implications of this finding are not adequately addressed in the discussion. See additional critiques below:
Intro:
Background discussion of TREM-1 is quite confusing particularly in reference to the Carneiro et al. paper where it is unclear how TREM-1 expression/signaling was studied in L. braziliensis-infected PBMCs. It is important that the conclusions from this study are clear as this lays basis for the current study.
Results:
Characteristics of study population were not well described and the referenced article does not include this info. There were clearly differences in the populations of infected and healthy controls from what little is reported and thus there may be a need to do propensity matching for a proper comparison.
Table 1 appears to be repeated in the manuscript document under the table 1 and table 2 headings
First line in subheading 2.2 needs a citation.
It seems inappropriate to pool healthy controls and infected patients for final analysis of cytokine production as these populations are quite different in their cytokine profiles.
Discussion:
Paragraph 1 is overly confusing, unclear what is the relevance of citing so many statistics to the results of this paper.
Discussion is overly long and mainly functions as a review of the literature, repeating some elements already addressed in the intro. There needs to be more critical discussion of why TREM-1 expression seems to be no different in infected and healthy cohorts.
Supplementary:
Units are never listed in the table
Author Response
Dear Professor,
We appreciate very much your comments for improving our manuscript. We have tried to reply to your comments to the best of our capacity.
Introduction
We rephrased the sentences Line 72 -77
CCL7, IL8, IFI44L and IL1B genes are upregulated in L. braziliensis-stimulated peripheral blood mononuclear cells (PBMCs) from healthy high IFN-γ producers individuals [15]. The upregulation was cited to be related to IL-17 and TREM-1 pathways by using Ingenuity pathway analysis [15]. In L. braziliensis-infected patients with CL, miR-193b, miR-671 and TREM-1 were shown to be negatively correlated with good response to treatment and faster wound healing [16].
Results
We have included in the text Line 96-100. We do agree with you that the controls are older than patients. We think that this shows that they are really resistant to Leishmania-infection as they have been exposed more in that environment. Furthermore, the genes we are studying are not fatal so that we can miss some genotypes. Of note, HWE is not deviated in both controls and patients. In addition all the P values were controlled by sex and age.
Line 96-100
All patients with CL included in the study had fewer or equal (≤) to six skin lesions. Patients with HIV were excluded. Pregnant women were also excluded. Diameters of the lesions were not recorded as all patients sought treatment only after the lesions are well developed. Besides most of the patients often applied local traditional treatment before going to hospital.
Table 1 appears to be repeated in the manuscript document under the table 1 and table 2 headings
Thank you pointing this error
We have corrected.
Discussion:
Paragraph 1 is overly confusing, unclear what is the relevance of citing so many statistics to the results of this paper.
We have eliminated Paragraph 1
Discussion is overly long and mainly functions as a review of the literature, repeating some elements already addressed in the intro. There needs to be more critical discussion of why TREM-1 expression seems to be no different in infected and healthy cohorts
We have tried to be more concised.
Reviewer 2 Report
In this manuscript, do Espírito Santo Júnior and colleagues describe the impact of the TREM1 rs2234237 mutation in patients infected with Leishmania guyanensis. The authors show that the mutation does not increase risk to the disease, and does not affect the quantities of circulating cytokines (at least in the population that the authors studied).
Although the study is interesting, the numerical data clearly show that there is no effect by the TREM1 SNPs on all cytokines, including IL-10 (even if the statistics reveal a very modest impact). To avoid misleading readers, this should be stated more clearly in both the abstract and discussion/conclusions section of the manuscript.
Having said this, this paper is effectively a report on otherwise negative results. To improve on the significance and citation potential of this study, the authors must show any clinical data that could point to a connection between TREM1 SNPs and IL-10 levels. Do the authors have histological data showing Giemsa-stained biopsies coming from patients with the various TREM1 SNPs? Do these show more/less infected phagocytes, or a difference in inflammatory infiltrates? Is there any mouse data?
Aside from the requested data, the manuscript should also be thoroughly edited. For example, the abstract contains repeated information in lines 36-37 and 40-41.
Author Response
Dear Professor,
Thank you very much for your suggestions to improve our manuscript. we apppreciate very much for dedicating som times to our manuscript and helping us with your comments.
We have tried to reply to your comments to the best of our understanding.
Although the study is interesting, the numerical data clearly show that there is no effect by the TREM1 SNPs on all cytokines, including IL-10 (even if the statistics reveal a very modest impact). To avoid misleading readers, this should be stated more clearly in both the abstract and discussion/conclusions section of the manuscript.
We have included in the text
Line 29 -30 (abstract)
The rs2234237 shows only a modest effect on plasma IL-10 that disappears when correction of p-value is applied.
line 163-166 (discussion)
In contrast, the variant may have an influence on the expression of IL-10 as the T allele correlated with lower levels of circulating plasma IL-10 although the statistical significance disappeared when corrections are applied for multiple comparisons.
Having said this, this paper is effectively a report on otherwise negative results. To improve on the significance and citation potential of this study, the authors must show any clinical data that could point to a connection between TREM1 SNPs and IL-10 levels. Do the authors have histological data showing Giemsa-stained biopsies coming from patients with the various TREM1 SNPs? Do these show more/less infected phagocytes, or a difference in inflammatory infiltrates? Is there any mouse data?
We agree with you but unfortunately we do not have histopathological data for this.
Aside from the requested data, the manuscript should also be thoroughly edited. For example, the abstract contains repeated information in lines 36-37 and 40-41.
Thank you for reminding us. We brought the correction.
Sincerely
Ramasawmy
On behalf of the authors
Reviewer 3 Report
In this manuscript, the authors analyze a cohort of individuals with cutaneous leishmaniasis (CL) and healthy controls to determine if a SNP (rs2234237 A/T- Thr25Ser) in the protein TREM1 predisposed individuals to development of CL and if it impacted the cytokine response. This SNP in TREM1 have been associated with a variety of diseases including sepsis, pneumonia, severe malaria, and IBD. The cohorts were predominantly male, but the ratio was similar between the groups and the healthy controls were older than the cases. The frequencies of each genotype and alleles were similar between the groups indicating that this SNP does not impact the development of CL. The authors then went on to evaluate difference in pro-inflammatory cytokines and chemokines in the presence or absence of this SNP. This comparison was done between CL and healthy controls and among specific genotypes using difference dominant and recessive modeling. While there was not much difference in the levels of IL-1b, IL-6, IL-8, MCP-1, and TNF-a in the plasma samples of each group, the presence of the T allele resulted in reduced levels of circulating IL-10. This was unrelated to L. guyanensis infection, as differences were only observed in when the groups were combined. Overall, the results support that this SNP does not largely impact L. guyanensis infection or cytokine production, but it does provide some potential insight into the biological changes of having this SNP.
Major comments:
- Was there a difference in the CL disease severity, such as lesion size, persistence, or other systems, in the cases between each genotype?
- Please include when the blood samples collected for the CL case group? Were they all drawn around the same time post-onset of symptoms or was another standardization point used?
- In supplementary table 1, the q-values are very high across all of the samples, including the IL-10 data. This is not consistent with the p value and corrected p value results, particularly for the IL-10 data. The authors should double check that this is indeed the q-values.
Minor comments:
- Please include more information about the impact of the rs2234237 A/T SNP in terms of TREM1 functionality. The authors mention where the mutation is located, but not if it impacts the activity of TREM1, binding partners, etc.
- Table 1 is repeated in the text. Please remove the duplicate.
- Line 110- should be “did not show any statistical difference”.
- The authors compare various models of genotypes including codominant, dominant, recessive, etc. In the severe malaria and Behcet’s disease, was this difference observed as codominant, dominant, recessive, or overdominant genotypes? Please discuss.
- Please indicate if the mean or median is shown for Figure 1 and Figures S1 and S2.
- Please include figure legends for Figures S1 and S2.
Author Response
Dear Professor,
We thank you very much for your positive comments to improve our manuscript. We are replying to your querries to the best of our knowledge.
Was there a difference in the CL disease severity, such as lesion size, persistence, or other systems, in the cases between each genotype
Unfortunately, we dot have these datas because most of the patients seek for treatment only when the lesions are well developed. The patients also applied local traditional treatment which make it difficult to evaluate. That is why we just coss genotypes to phenotypes. By the way we have included in the text the reason
line 96-100
All patients with CL included in the study had fewer or equal (≤) to six skin lesions. Patients with HIV were excluded. Pregnant women were also excluded. Diameters of the lesions were not recorded as all patients sought treatment only after the lesions are well developed. Besides most of the patients often applied local traditional treatment before going to hospital.
Please include when the blood samples collected for the CL case group? Were they all drawn around the same time post-onset of symptoms or was another standardization point used?
We have included in the text
Line 260-261
At the time of enrolment in the study, all participants provided 5 mL of peripheral blood. All patients with CL were treatment naïve prior to collection of blood
In supplementary table 1, the q-values are very high across all of the samples, including the IL-10 data. This is not consistent with the p value and corrected p value results, particularly for the IL-10 data. The authors should double check that this is indeed the q-values.
Our statisticians double check the data and confirm the p and q values.
Minor comments
Please include more information about the impact of the rs2234237 A/T SNP in terms of TREM1 functionality. The authors mention where the mutation is located, but not if it impacts the activity of TREM1, binding partners, etc
To date, this polymorphism has only been correlated to soluble TREM
Table 1 is repeated in the text. Please remove the duplicate.
We have corrected.
Line 110- should be “did not show any statistical difference”.
We have corrected.
The authors compare various models of genotypes including codominant, dominant, recessive, etc. In the severe malaria and Behcet’s disease, was this difference observed as codominant, dominant, recessive, or overdominant genotypes? Please discuss
We have included in the text the discussion line 201-217
Comparisons of genotypes frequencies did not reveal any statistical differences between the patients with CL and HC, but point out that carriers of the T allele may have 18% risk of developing CL, in a dominant model (OR=1.18 (95%CI [0.95 – 1.47]). In a study of falciparum malaria with 293 and 87 African children with uncomplicated and severe malaria respectively, individuals with the T allele had increased risk of developing severe malaria (dominant model; OR= 2.4 (95%CI [1.2 – 4.5]) [26]). Likewise, under dominant model, the T allele was associated with BD (OR= 1.69 (95%CI [1.1 – 2.6]) [21] Although no statistical significance has been observed in our analysis, we can just hypothesize that individuals with the T allele may not control properly the inflammation and just progress to the development of lesion. The rs2234237 T allele was cited to correlate with an increased level of soluble TREM-1 in human whole blood stimulated with LPS [25]. Soluble TREM-1 (sTREM-1) may act as a decoy receptor for TREM-1 ligands and leads to a decrease of circulating IL-10. Interestingly, TREM-1 silencing lowered the expression of IL-10 [34]. Contrastly, sTREM-1 was suggested as a possible modulator of anti-inflammatory pathway [37]. However, it is necessary to have more studies with larger samples-size of patient with CL to look at the influence of this variant on plasma IL-10 levels and its involvement in the pathogenesis of CL.
Please indicate if the mean or median is shown for Figure 1 and Figures S1 and S2.
We have indicated in the figures.
Please include figure legends for Figures S1 and S2.
Thank you for reminding us this omission.
We hope we have been able to answer part of your querries.
Once again, We thank you very much for your comments.
Sincerely
Ramasawmy on behalf of all the authors
Round 2
Reviewer 1 Report
The revised form of the manuscript still does not address several major shortcomings of the study. Chief among these is the pooling of healthy controls with infected cases for analysis of cytokine expression. The infected cases have much higher cytokine levels throughout and are clearly not equivalent to their matched healthy control counterparts in this regard. The fact that these needed to be pooled to show any difference in just one cytokine actually argues that there is no impact of this polymorphism on cytokine expression unless extreme manipulations are made to the data. In general this paper's results clearly show that the polymorphism has no impact on acquisition of leishmaniasis or cytokine expression as a whole, and this should be reflected in the discussion. It is not clear from the introduction or discussion why this particular TREM1 polymorphism is of any interest, and what the expectation of its impact in leishmaniasis and other infectious and inflammatory conditions might be.
Author Response
Dear Reviewer,
First of all, we would like to thank you for helping us to improve the quality of the manuscripts with your comments.
Querry 1
Chief among these is the pooling of healthy controls with infected cases for analysis of cytokine expression. The infected cases have much higher cytokine levels throughout and are clearly not equivalent to their matched healthy control counterparts in this regard. The fact that these needed to be pooled to show any difference in just one cytokine actually argues that there is no impact of this polymorphism on cytokine expression unless extreme manipulations are made to the data. In general this paper's results clearly show that the polymorphism has no impact on acquisition of leishmaniasis or cytokine expression as a whole, and this should be reflected in the discussion.
We do understand your worry about the pooling of the patients and controls to show a modest impact on plasma IL-10. As we are looking at whether the variant has an influence on expression, we thought that we can do the pooling. While it is true that the patients have higher plasma cytokines than the controls and the controls are older. By linear regression correlation analysis, we showed that age and gender have no influence on plasma IL-10. We have included a supplementary figure. Furthermore, we have included a paragraph about this limitation in the discussion section highlighted in yellow (Line 215-225).
Querry 2
It is not clear from the introduction or discussion why this particular TREM1 polymorphism is of any interest, and what the expectation of its impact in leishmaniasis and other infectious and inflammatory conditions might be.
We have in the text (Line 87: Furthermore, this variant is correlated with increased level of soluble TREM-1 [25].
Line 88 Taken into account the importance of TREM-1 in inflammatory diseases
Reviewer 2 Report
The revised submission shows an overall improvement. It is too bad that the authors could not provide supporting data.
Author Response
Dear reviewer,
We appreciate very much your comments and happy to hear that you have observed an overall improvement.
The revised submission shows an overall improvement. It is too bad that the authors could not provide supporting data.
We also regret that we do not have these data. However, we think that this article can be a lead for other group of researchers in immunogenetics of Leishmaniasis.
Once again, we apprciate very your precious comments for helping us to improve the manuscript.
Sincerely,
Ramasawmy